# Ruminal Bacterial Community Successions in Response to Monensin Supplementation in Goats

**DOI:** 10.3390/ani12172291

**Published:** 2022-09-04

**Authors:** Xi Guo, Yuqin Liu, Yu Jiang, Junhu Yao, Zongjun Li

**Affiliations:** College of Animal Science and Technology, Northwest A&F University, Yangling, Xianyang 712100, China

**Keywords:** ruminal bacterial community, temporal dynamics, monensin supplementation, goats

## Abstract

**Simple Summary:**

Monensin has been successfully used in the ruminants’ diets to manipulate ruminal fermentation and improve feed efficiency, but its use is facing decreased levels of social acceptance due to the potential impacts on public health. Understanding the ruminal bacterial community successions in response to monensin supplementation would help the search for alternatives. We found that the ruminal ecosystem was reshaped through a series of succession processes during the adaption to monensin rather than following a clear dichotomy between Gram-positive and Gram-negative cell types, and the carbohydrate-degrading bacteria presented a higher adaptability. Therefore, a potential alternative for monensin as a rumen modifier could be one with similar patterns of ruminal microbial community successions.

**Abstract:**

Previous studies have demonstrated that the effects of monensin on methanogenesis and ruminal fermentation in ruminants were time-dependent. To elucidate the underlying mechanism, we investigated the ruminal bacterial community successions during the adaptation to monensin supplementation and subsequent withdrawal in goats. The experiment included a baseline period of 20 days followed by a treatment period of 55 days with 32 mg monensin/d and a washout period of 15 days. Monensin supplementation reduced the α diversity and changed the structure of ruminal microflora. The α diversity was gradually restored during adaption, but the structure was still reshaped. The temporal dynamics of 261 treatment- and/or time-associated ruminal bacteria displayed six patterns, with two as monensin-sensitive and four as monensin-resistant. The monensin sensitivity and resistance of microbes do not follow a clear dichotomy between Gram-positive and Gram-negative cell types. Moreover, the temporal dynamic patterns of different bacterial species within the same genus or family also displayed variation. Of note, the relative abundance of the total ruminal cellulolytic bacteria gradually increased following monensin treatment, and that of the total amylolytic bacteria were increased by monensin, independent of the duration. In conclusion, under the pressure of monensin, the ruminal ecosystem was reshaped through a series of succession processes, and the carbohydrate-degrading bacteria presented a higher level of adaptability.

## 1. Introduction

Ruminants ultimately convert human-indigestible plant resources into human-digestible milk or meat products, and the bioconversion efficiency depends on ruminal microbiota [1]. Manipulating ruminal microbiota to increase an animal’s feed efficiency has received widespread attention since feed cost could account for 50–70% of the gross expenditure in the ruminant industry [2]. Monensin, as an ionophore antimicrobial feed additive, has been successfully used in the diets of domestic ruminants to manipulate ruminal fermentation and improve feed efficiency for decades [3,4,5]. However, its use is facing decreased levels of social acceptance due to the potential impacts on public health [6,7] and has been banned in the European Union and limited in China. Recently, many works have been conducted to search for potential alternatives for monensin [8,9,10,11,12].

Studies using pure cultures of bacteria suggested that ruminal Gram-positive bacteria are more sensitive to monensin than Gram-negative species [13,14]. However, mounting evidence suggests that the regulation mechanism of monensin is more complex than previously believed [15]; specifically, the cell wall model of monensin sensitivity and resistance does not always follow a clear dichotomy [13,16]. For example, some taxa of ruminal Gram-positive bacteria (e.g., Veillonellaceae) were resistant to monensin and some taxa of Gram-negative bacteria (e.g., Succiniclasticum) were sensitive [17,18,19]. Beyond that, the reduction of ruminal protozoa populations recovered under long term monensin supplementation in cows [20], and the weakening of postprandial responses to monensin as reflected by changes to the rumen fermentation parameters likely reflects the development of resistance to monensin by ruminal microbes [20,21]. However, the dynamic patterns of the rumen bacterial community during the adaptation to monensin remained unclear. We hypothesized that understanding the underlying mechanisms of monensin on rumen manipulation would help in the search for alternatives, as alternatives with similar patterns of ruminal microbial community successions to monensin would have more potential. Therefore, the objective of the present study was to trace the ruminal bacterial community successions during the adaptation to monensin supplementation and the subsequent withdrawal in goats via the amplicon sequencing of the 16S rRNA gene.

## 2. Materials and Methods

### 2.1. Experimental Design, Sample Preparation and 16S rRNA Sequencing 

The experimental design and animal feeding procedure have been described previously [21] with minor modifications. Briefly, five non-lactating Xinong Saanen dairy goats (4 years of age) with permanent ruminal cannulas and similar BW (54 ± 2.4 kg) were used in this study. The goats were housed in temperature-controlled chambers (6–14 °C throughout the experiment) and limit-fed (concentrate: forage, 40:60, Appendix A) to reduce the confounding factors of ambient temperature and feed intake, respectively. The goats were healthy and had never been exposed to monensin prior to the experiment. The feeding experiment lasted for 90 days, including a baseline period (without monensin) of 20 days, a treatment period (32 g monensin per day, top dressed) of 55 days, and a washout period (monensin withdrawal) of 15 days. The animals were fed the same amount of the total mixed ration, equivalent to 408 g of concentrate, 281 g of corn silage, and 330 g of alfalfa hay, twice daily at 2 equal meals at 0800 and 1800 h. The amounts of feed refused, if any, were recorded daily. On average, >97% of the diet offered daily to each goat was consumed, and no difference was observed in the dry matter intake between treatments or periods. The goats had free access to drinking water.

Rumen samples were collected on the last day of the baseline period and washout period, and on d 10, d 30, and d 50 of the treatment period. On the day of sample collection, rumen content samples of each goat were collected (about 40 mL) from the anterior ventral sac of the rumen at 0, 1, 2, 4, 6, and 8 h after the morning feeding and then strained through 4 layers of cheesecloth. The six collected rumen fluid samples for one day from each goat were composited equally into one sample and then centrifugated by 500× *g* for 5 min to remove feed particles and protozoa. The 200 µL supernatant was subsampled to extract the total DNA using the QIAamp DNA Stool Mini Kit (Qiagen, Hilden, Germany) according to the manufacturer’s instructions. The density and purification of the extracted DNA were detected using a Nanodrop spectrophotometer (Thermo Fisher Scientific, Inc., Madison, WI, USA). The V4 hypervariable region of the bacterial 16S rRNA gene was amplified using the primers 520F (5′-AYTGGGYDTAAAGNG-3′) and 802R (5′-TACNVGGGTATCTAATCC-3′). The PCR reaction conditions were: 94 °C for 5 min; 94 °C for 30 s, 50 °C for 30 s, and 72 °C for extension; repeated for 27 cycles; with a final 72 °C for 7 min. The PCR products were excised from a 2% agarose gel and purified using an AxyPrep DNA Gel Extraction Kit (Axygen Biosciences, Foster City, CA, USA). Then amplicon libraries were constructed and sequenced (250 bp paired-end reads) using an Illumina MiSeq platform.

Original pair-end sequences with a sequence length shorter than 150 bp, a mean quality lower than 30, adaptor contamination, ambiguous bases, or host-contaminating reads were removed as described by Ding et al. (2017) [22]. The quality-filtered sequence reads that contained > 10 bp sequence overlaps without any mismatch were assembled into trimmed sequences according to their overlap sequence. The trimmed sequences were clustered into operational taxonomic units (OTUs) at ≥97% sequence similarity using Uclust in QIIME [23]. Subsequently, the taxonomy of OTUs were assigned using the Greengenes databases [24]. OTUs with at least 15 sequences in the total samples were retained for further analysis.

### 2.2. Statistical Analysis

The alpha diversity of the samples was estimated using the abundance-based coverage (ACE) estimators (community richness), Shannon indices (community diversity) and observed OTUs. The Bray–Curtis distance [25] on the relative abundance of OTUs across samples was computed and visualized through principal coordinate analysis (PCoA) plots in R v.3.6.0 (http://www.R-project.org; accessed on 15 January 2021). A permutational multivariate analysis of variance (PERMANOVA) was performed using the adonis function in the R package vegan (https://github.com/vegandevs/vegan/; accessed on 20 January 2021) to compare the statistical difference of microbial composition across the experimental periods and between any two phases. Considering that the susceptibility of bacteria to monensin is mainly cell-wall dependent [26] and that bacterial Gram stain can be mostly verified at the family level [27,28], a PCoA was also conducted for the dominant families (average relative abundance >0.01%).

To identify the relative abundance changes of microbial OTUs across the experimental periods, student t-tests for paired-sample differential analyses were performed using the R packages EdgeR [29] with an animal as a random effect. The dynamic patterns of differential OTUs (*p* < 0.05) between any pair of experimental periods were clustered into different modules using an R package pheatmap in an unsupervised way. In each cluster, the relative abundance of OTUs within the same cluster were normalized and the curve-fitting regression line was generated using the Loess curve-fitting method in R. To better understand the relationship between the temporal dynamic patterns and phylogenetic profile, the sequences of time- or treatment-associated OTUs within the genus Prevotella were aligned using MAFFT v. 7.407 [30], and then maximum likelihood trees were constructed using IQ-TREE v. 1.6.9 [31].

Student *t*-tests for paired samples were also performed to test differences across the experimental periods for alpha diversity indices and the relative abundance of functional microbial groups (dominant family, amylolytic, and cellulolytic bacteria). The probability levels were set at *p* < 0.05 for significance and at 0.05 ≤ *p* < 0.10 for a trend.

## 3. Results

### 3.1. Bacterial Community Composition and Diversity

After concatenation and quality filtering, a total of 1,521,422 sequences (60,857 per sample) were obtained from the 25 samples. According to the clustering at the 97% similarity level, 2365 OTUs were generated, of which 71.4% were shared across the 5 time points (Appendix A). The OTUs were assigned to 22 phyla, 33 classes, 57 orders, 70 families, and 69 genera. Compared with the baseline period, the number of observed OTUs and ACE indices decreased at d 10 of monensin treatment (*p* = 0.056 and 0.028, respectively), and gradually restored over time, up to the pre-supplementation level on d 30 of treatment (Figure 1). No difference was observed for Shannon indices. 

Principal coordinate analysis (PCoA) (Figure 2 and Table 1) revealed that the ruminal bacterial community structures varied significantly across phases at both the OTU level (Bray-Curtis RANOSIM = 0.345, *p* = 0.001) and family level (Bray-Curtis RANOSIM = 0.154, *p* = 0.032). The between-phase ANOSIM (Figure 2 and Table 1) results showed that the bacterial community structures in the treatment period were different than those in the baseline period, with a maximum difference on d 10. At the OTU level, a more heterogeneous microbiota occurred on d 10 and then developed into a more homogeneous microbiota during the treatment period. The bacterial community structures between the baseline period and washout period were no different at the family level, but were different at the OTU level (Bray-Curtis RANOSIM = 0.716, *p* = 0.011).

### 3.2. Succession Patterns of Bacterial Communities

A total of 261 OTUs that had differentiated at least once (*p* <0.05) in relative abundance between any two phases (Appendix A) were tagged as treatment- and/or time-associated. For an unsupervised hierarchical clustering analysis, we clustered the 261 OTUs into six dynamic patterns of microbial succession (defined as C1–C6) (Appendix A and Figure 3). The relative abundance of microbes in C1 (n = 62) and C2 (n = 36) decreased after monensin introduction, while those in C5 (n = 28) and C6 (n = 21) increased after monensin introduction and continuously increased during the adaptation to monensin. After the monensin withdrawal, the relative abundance of microbes in C2 and C5 recovered to the pre-supplementation level, while those in C1 and C6 did not, presenting aftereffects to monensin. In C3 (n = 79) and C4 (n = 35), the relative abundance of microbes reached a peak on d 10 and d 30 of the treatment period, respectively, and then returned to the pre-supplementation level. We classified the bacteria in C1 and C2 as monensin-sensitive microbes and the others as monensin-resistant microbes.

The OTUs in both Gram-positive and Gram-negative bacteria, even in the same genus or family (e.g., Prevotellaceae and BS11), displayed all of the six temporal dynamic patterns over the course of the experiment (Figure 4A). Among the 231 treatment- and/or time-associated Gram-negative bacteria, 89 (38.5%) presented remarkable monensin sensitivity, the proportion was close to that in Gram-positive bacteria (37.5%). No monensin-sensitive bacteria were found in family S24-7 (order Bacteroidales, Gram-negative), which increased in its relative abundance by three-fold (*p <* 0.01) at d 10 of the monensin treatment and gradually restored to the pre-supplementation level on d 50 of the treatment (Figure 4A and Appendix A). The family Prevotellaceae (order Bacteroidales, Gram-negative) also presented a high resistance to monensin (Figure 4A and Appendix A); its relative abundance rose (*p* < 0.05) by over one-fold, exceeding that of family *BS11* (order Bacteroidales) as the most predominant family (Appendix A). On the other hand, the relative abundance of family RF16, also in the order Bacteroidales, decreased (*p* < 0.01) by 72.1% on d 10 of the monensin treatment, and the proportion of monensin-sensitive bacteria increased up to 84.6% (11/13, Figure 4A). Moreover, the bacteria in the genus Prevotella with the same temporal dynamic patterns did not cluster in the phylogenetic tree (Figure 4B).

### 3.3. Succession Patterns of Amylolytic and Cellulolytic Genera

The relative abundance of the total amylolytic bacteria genera (Prevotella, Bifidobacterium, [Prevotella], Ruminobacter, Selenomonas, and Succinivibrio) were significantly increased by monensin supplementation, independently of the suppl’mentation’s duration (Figure 5). After the monensin withdrawal, the relative abundance of the total amylolytic bacteria were reduced but still higher than that of the total at the baseline period. The relative abundance of the total cellulolytic bacteria genera (Butyrivibrio, Clostridium, Fibrobacter, and Ruminococcus) were increased following monensin treatment, and it became significantly higher (*p* = 0.049) on d 50 than that at baseline period (Figure 5).

## 4. Discussion

Understanding the regular patterns of change in the community structure over time is a fundamental pursuit of ecology [32,33]. Monensin is one of the most effective feed additives in terms of improving feed efficiency by regulating rumen fermentation [3,4]. Thus, the current study aims to track its reshaping process in rumen microbial communities. The results showed that the effect of monensin on rumen bacterial alpha diversity was time-dependent and that the reduction in the alpha diversity was restored after 30 days of treatment. This result explains the different responses between two previous studies, in which the bacterial diversity within the rumen was reduced at 20 days of monensin treatment in lactating dairy cattle [34] while remaining unchanged at 60 days of monensin treatment in fattening lambs [35]. At the OTU level, the heterogeneous microbiota on d 10 of monensin treatment suggested the beginning of the development of a rumen bacterial community with large fluctuations, which then developed into a more convergent and stable microbiota during the treatment period. However, a small part of the bacteria (in C5 and C6) did not stabilize after 50 days of adaption, suggesting that establishing a new microbial population balance through monensin supplementation takes longer than 2 weeks, as previously described [36]. The adaptive duration for establishing a new rumen microbial population balance even lasted more than 12 wk after the complete removal of ruminal protozoa [37]. Therefore, the long-term dynamical effects of a dietary treatment in animal nutritional research should be an important index for systematic evaluation rather than short-term effects with one time-point.

In terms of the mechanism of action, monensin has been proposed to preferentially suppress Gram-positive bacteria [13,14] without an outer lipopolysaccharide layer to limit the penetration of monensin to the cell membrane [13]. A novel finding of the present study noted that the temporal dynamic patterns of ruminal bacteria during adaptation to monensin do not follow a clear dichotomy between Gram-positive and Gram-negative cell types. Interestingly, the proportion of Gram-negative bacteria among the observed remarkable monensin-sensitive bacteria was much higher than that of Gram-positive bacteria. Moreover, the developments of different bacterial OTUs within the same genus or family also displayed different, and even opposite, trends. The variations in the monensin-resistance ability of 14 Prevotella strains were also observed in an in vitro study [38]. In conjunction with the poor relationship between temporal dynamic patterns and phylogenetic profile, these results suggest that the susceptibility of bacteria to monensin should be defined at the OTU level, and even more likely, at the strain level. These results also suggest that the susceptibility of bacteria to monensin is not only dependent on the cell-wall type, confirming the previous findings [16,19]. Thus, the conclusion drawn from this study provides additional evidence and support for updating the textbook understanding on bacterial monensin sensitivity but also uncovers more complexity in its underlying mechanism. More molecular evidence and further study is needed.

With respect to temporal dynamic patterns of monensin-insensitive bacteria, their complex succession patterns under the selective pressure of monensin are most likely explained by the nutrient-niche competition. After the introduction of monensin, a group of rumen bacteria (in C3) quickly occupied the niche of monensin-sensitive microbes, but only 32.9% of these bacteria retained the niche’s advantages until d 30 of the monensin treatment (Appendix A). About 65% of the quickly rising bacteria were in the families Prevotellaceae, [Paraprevotellaceae], BS11, and S24-7, which are rumen-predominant (52.6% in this study) amylolytic bacteria [39,40,41,42]. A possible reason for the increased level of amylolytic bacteria could be their functional redundancy with rumen protozoa in the starch nutrient niche; and the dynamic patterns of the rumen protozoal community during the adaptation to monensin were potentially opposite of that of the quickly rising bacteria, although this was not tracked in the present study. The inhibitive effects of monensin on protozoa is widely accepted [3,13,43]; however, protozoa adapt to monensin by changing their membrane structure [43]. Guan et al. (2006) [20] observed a reduction of ruminal protozoal populations in cows receiving monensin, but that reduction appeared to be a short-term effect, with a complete recovery after 4 to 6 wk of treatment. Following the drop of the quickly rising bacteria, the second and third groups of rumen bacteria rose in succession on d 30 and d 50 of the monensin treatment, respectively, likely because these groups obtained a higher niche-competitive advantage or monensin-resistance. Of note, the relative abundance of the total ruminal cellulolytic bacteria gradually increased following monensin treatment, which likely explains the previous result in which the monensin-induced reduction of alfalfa degradation in goats faded within days of the treatment [21]. Taken together, the feed carbohydrate-degrading bacteria likely presented higher niche-competitive advantages in the rumen community under the selective pressure of monensin. After the monensin withdrawal, the relative abundance of most treatment-associated bacteria returned to the pre-supplementation level, while aftereffects of monensin were observed in 83 OTUs (in C1 and C6), suggesting a strong selection pressure for these microbes.

## 5. Conclusions

To our knowledge, this investigation is the first to study the ruminal bacterial community successions in response to monensin supplementation using 16S rRNA Sequencing. In conclusion, under the selective pressure of monensin, the ruminal ecosystem was reshaped through a series of succession processes, and the feed carbohydrate-degrading bacteria presented a higher adaptability. The temporal dynamic patterns of ruminal bacteria during the adaptation to monensin could partially elucidate our previous result in ruminal fermentation and feed degradation. Further research is needed to fully understand the molecular and evolutionary mechanisms underlying the succession difference of rumen bacteria or protozoa during the adaptation to monensin using their genomes [42,44].

## Figures and Tables

**Figure 1 animals-12-02291-f001:**
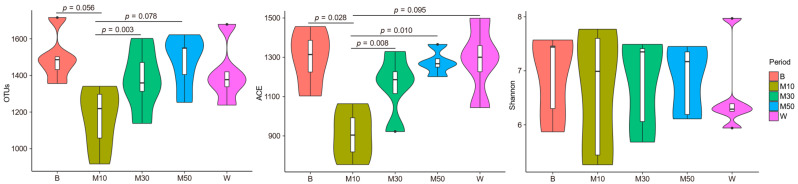
Dynamics of the observed OTUs and α diversity of the ruminal bacterial community following monensin treatment in goats. Note: B, baseline period; M10, M30, and M50, on d 10, 30, and 50 of the treatment period; W, washout period; *p*, *p*-value.

**Figure 2 animals-12-02291-f002:**
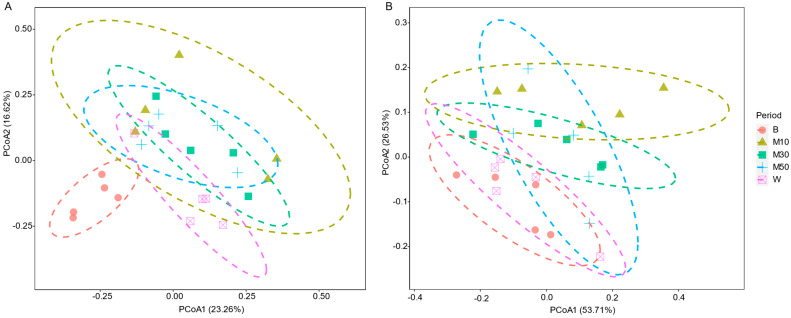
Bray-Curtis distance principal coordinate analysis (PCoA) of ruminal bacterial community successions following monensin treatment at the OTU level (**A**) and at the family level (**B**). The ellipses around each treatment group are of 80% confidence. Note: B, baseline period; M10, M30, and M50, on d 10, 30, and 50 of the treatment period; W, washout period.

**Figure 3 animals-12-02291-f003:**
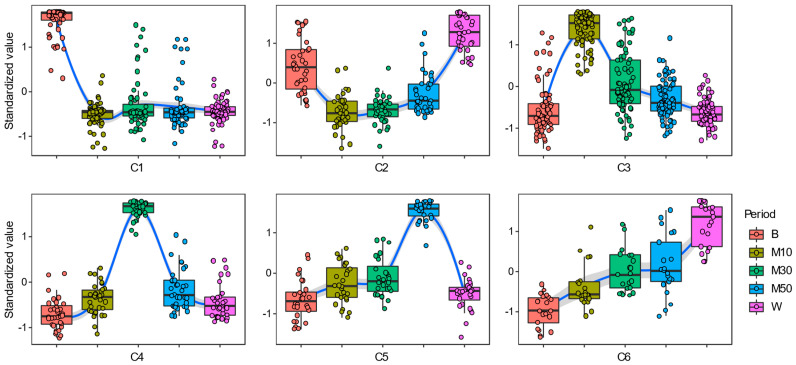
Six temporal dynamic patterns (C1–C6) of ruminal bacteria during the adaptation to monensin in goats. Note: B, baseline period; M10, M30, and M50, on d 10, 30, and 50 of the treatment period; W, washout period.

**Figure 4 animals-12-02291-f004:**
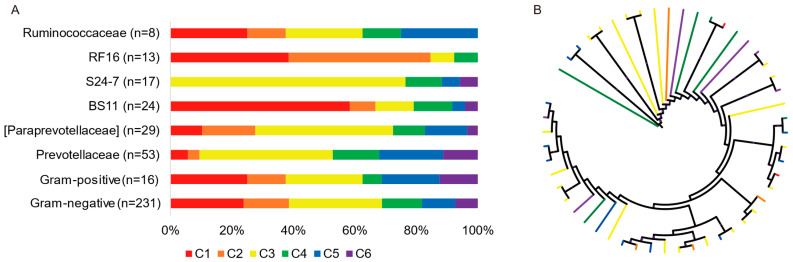
The proportion of the six temporal dynamic patterns in dominant families (average relative abundance > 1%) and Gram-positive or negative bacteria (**A**). The phylogenetic tree of the treatment- and/or time-associated OTUs within the genus Prevotella (**B**).

**Figure 5 animals-12-02291-f005:**
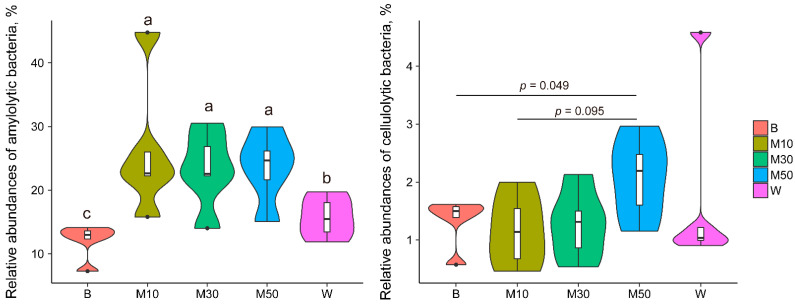
Dynamics of the amylolytic and cellulolytic bacteria following the monensin treatment in goats. Note: B, baseline period; M10, M30, and M50, on d 10, 30, and 50 of the treatment period; W, washout period. Different letters indicate significant difference between groups. Different lowercase letters between periods indicate a significant difference (*p* < 0.05).

**Table 1 animals-12-02291-t001:** Permutational multivariate analysis of variance of the microbial diversity among different phases.

	OTUs		Family	
	*R*	*p*	*R*	*p*
Total	0.345	0.001	0.154	0.032
B vs. M10	0.664	0.012	0.524	0.007
B vs. M30	0.848	0.006	0.304	0.079
B vs. M50	0.816	0.009	0.392	0.025
B vs. W	0.716	0.011	0.000	0.412
M10 vs. W	0.244	0.055	0.368	0.024

*R* value indicates the mean rank of between group dissimilarities; *p* value indicates whether the difference between different groups is significant.

## Data Availability

All sequencing data in this study have been deposited in the NCBI database with the accession ID PRJNA853530.

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
