# Peer review of "Ruminal Bacterial Community Successions in Response to Monensin Supplementation in Goats"

_animals, 2022, doi:10.3390/ani12172291_

Round 1

Reviewer 1 Report

The study describes microbial changes, examined at molecular level when goats are being supplemented with monensin.  The study has some novel aspects, and appears to be executed at appropriate level, at least at laboratory analysis level.  However, there are some apparent gaps,  For example, why did authors choose to study this in goast, as opposed some main ruminant categories normally targeted for monensin implementation?   Only six animals were used - how is this justified?

The Introduction needs to outline significance of this research.  Why it is important that we know mode of action of monensin? Aim and hypothesis are missing, making it difficult to understand the whole study and the discussion points. The overall writing needs to be improved.  Also there is a fair bit of information presented in figures, authors need to revise and see if all of these are needed.

Some specific comments listed below

L45-46 and L205 – I disagree with this statement, either delete or provide evidence that others are not as effective

L57-58 – This is methodology and does not belong here.  Proper aims need to be spelled out

L231 – What is

Author Response

The study describes microbial changes, examined at molecular level when goats are being supplemented with monensin.  The study has some novel aspects, and appears to be executed at appropriate level, at least at laboratory analysis level.

Response: Thank you for recognizing the potential contribution of this manuscript. We also appreciate your constructive comments and suggestions, which helped us in revising the manuscript. We revised the MS by incorporating your comments and the comments from the editor and the other reviewer. Please see the details as below regarding how we have addressed each of your comments and suggestions. We hope that your comments and concerns have been properly addressed.

However, there are some apparent gaps,  

For example, why did authors choose to study this in goats, as opposed some main ruminant categories normally targeted for monensin implementation?

Response: Monensin has been widely used in the beef cattle and dairy cattle in North American; however, it has been limited, particularly on lactating cow, in China (L42-44). Goats is an ideal alternative model due to its operability and sharing similar microbial community compositions with cattle. Moreover, goats are also important ruminant livestock, and the effect of monensin on goats was similar with that on cattle. In addition, and the province of our university located is the largest dairy goat raising province in China. Therefore, we chose dairy goats as our research animals in this study.

Only six animals were used - how is this justified?

Response: The rumen microbial composition commonly was individualized among animals (Xue et al., 2021); therefore, we used experimental design of repeated measures (self-comparison) to reduce the individualized fluctuations. Moreover, the Student t-test for paired samples with animal as a random effect would produce a high sensitivity result (Li et al., 2018). As unlike ANOVA, the Student t-test for paired-samples tests the null hypothesis that the differences of paired-samples is 0 and just one estimate needs to be made of the variance of populations’ differences (SED).

However, fluctuations in DMI and temperature during the entire experiment period maybe influence the rumen microbial composition. Therefore, we minimized the potential fluctuations by controlling the amount of TMR offered and maintaining a relatively stable temperature (6 to 14°C) in the environmentally controlled chambers (L76-78 ).

Li, Z., H. Ren, S. Liu, C. Cai, J. Han, F. Li, and J. Yao. 2018. Dynamics of methanogenesis, ruminal fermentation, and alfalfa degradation during adaptation to monensin supplementation in goats. Journal of Dairy Science 101:1048–1059. doi:10.3168/jds.2017-13254.

Xue, M.-Y., Y.-Y. Xie, Y.-F. Zhong, J.-X. Liu, L.L. Guan, and H.-Z. Sun. 2021. Ruminal resistome of dairy cattle is individualized and the resistotypes are associated with milking traits. anim microbiome 3:18. doi:10.1186/s42523-021-00081-9.

The Introduction needs to outline significance of this research.  Why it is important that we know mode of action of monensin?

Response: The potential alternative for monensin having closer patterns of ruminal microbial community successions to monensin would be more potential . This sentence was added in L65-67 in the revised MS.

Aim and hypothesis are missing, making it difficult to understand the whole study and the discussion points. The overall writing needs to be improved.  

Response: The aim and hypothesis were provided at the end of introduction (L62-70) in the revised MS.

Also there is a fair bit of information presented in figures, authors need to revise and see if all of these are needed.

Response: The Figure 5 was moved into the supplementary files (Fig S3).

Some specific comments listed below

 L45-46 and L205 – I disagree with this statement, either delete or provide evidence that others are not as effective

Response: The sentence in the L45-46 was deleted as suggested. While the sentence in the L205 was reworded (L 230).

L57-58 – This is methodology and does not belong here.  Proper aims need to be spelled out

Response: The sentence in the L57-58 was reworded (L67-70). The aim and hypothesis were provided at the end of introduction (L62-70 ) in the revised MS.

L231 – What is

Response:  “What’s more” was changed to “Moreover” (L256 ). 

Reviewer 2 Report

Xi Guo and coworkers present a study “Ruminal Bacterial Community Successions in Response to  Monensin Supplementation in Goats”. The manuscript is generally well written and presented data are of interest into the field.  However, both the summary and the conclusions should be improved in order to make the work more understandable and objective. Additionally, the objectives of the work would be better understandable if the line of investigation was better contextualized in the introduction. I believe that after a minor revised the paper is ready for acceptance

Author Response

Xi Guo and coworkers present a study “Ruminal Bacterial Community Successions in Response to Monensin Supplementation in Goats”. The manuscript is generally well written and presented data are of interest into the field.  However, both the summary and the conclusions should be improved in order to make the work more understandable and objective. Additionally, the objectives of the work would be better understandable if the line of investigation was better contextualized in the introduction. I believe that after a minor revised the paper is ready for acceptance.

Response: Thank you for recognizing the potential contribution of this manuscript. We also appreciate your constructive comments and suggestions, which helped us in revising the manuscript. We revised the MS by incorporating your comments and the comments from the editor and the other reviewer. We hope that your comments and concerns have been properly addressed.

The aim and hypothesis of this study were provided at the end of introduction (L62-70 ) in the revised MS.